# ContextKT: A Context-Based Method for Knowledge Tracing

Minghe Yu [1,*][ID], Fan Li [2][ID], Hengyu Liu [2][ID], Tiancheng Zhang [2][ID] and Ge Yu [2][ID]

1    Software College, Northeastern University, Shenyang 110169, China
2    School of Computer Science and Engineering, Northeastern University, Shenyang 110169, China
*    Correspondence: yuminghe@mail.neu.edu.cn

**Abstract:** Knowledge tracing, which is used to predict students' performance based on their previous practices, has attracted many researchers' attention. Especially in this rising period of intelligent education, many knowledge tracing methods have been developed. However, most of the existing knowledge tracing methods focus on the personality of practices and knowledge concepts but ignore the contexts related to the studying process. In this paper, we propose a context-based knowledge tracing model, which combines students' historical performance and their studying contexts during knowledge mastery. To be specific, we first define five studying contexts for performance prediction. The basic context is the current knowledge state of a student, which is described by their practice sequences. Then, a QR-matrix is defined to represent the relationship among questions, knowledge concepts, and responses, which describes the contexts of questions and knowledge. Furthermore, an improved LSTM model is proposed to capture the context of students' memory and forgetness, and a multi-head attention mechanism is designed to capture the context of students' behaviors. Finally, based on the captured contexts, the prediction model ContextKT is established. Our prediction model is evaluated on two real educational datasets. The experimental results show our model is effective and efficient in student performance prediction, and it outperforms the other existing methods.

**Keywords:** knowledge tracing; LSTM; attention mechanism; intelligence education

## 1. Introduction

With the development of intelligent education, many emerging computer techniques are combined with traditional education areas, and become an innovative research direction. To understand students' knowledge masteries and give them personalized supervising, knowledge tracing has attracted researchers' attention, which can model the knowledge states of students based on their previous practices. Knowledge tracing is a learner model that uses the responses to previous questions to predict the student's performance on the new questions.

Knowledge tracing is first proposed by Prof. John Anderson of Carnegie Mellon University to monitor students' changing knowledge state in skill acquisition of procedural knowledge [1]. According to the ACT-R theory of skill knowledge [2], skill knowledge is distinguished as declarative and procedural knowledge. Declarative knowledge is factual or experiential knowledge, for example, the concepts of C language like variable types, function definitions, and program structures, which can be acquired through experiences such as reading. Procedural knowledge is goal-oriented knowledge about problem-solving, for example, the procedural rules to code a C program to implement an algorithm, which can only be acquired by a sequence of practices by using declarative knowledge. At the time, knowledge tracing aims for two purposes: (1) to assist in predicting a student's performance in the mastery of knowledge accurately; (2) to assist in a student's practice sequence according to their performance prediction to enable the student to master the skill efficiently. Prof. John Anderson applied their knowledge tracing model in an intelligent programming tutor (APT) for students of universities or high schools to practice programming in Lisp, Prolog or Pascal.

Nowadays, with the help of new knowledge tracing techniques, people can develop a variety of applications [3–10] for intelligent education. First of all, it can be used in personalized resource recommendations. Based on the prediction result of knowledge tracing, instructors can capture the knowledge mastery of each individual student. Then they can recommend corresponding study resources or learning plans. Secondly, knowledge tracing can help students in adaptive learning. Knowledge tracing can help instructors understand students' study abilities to design or adjust proper teaching schemes for them. Different from resource recommendations that give students existing resources, adaptive learning redesigns a new scheme on established teaching contents. However, the two applications are similar in that both of them need to know students' knowledge masteries based on knowledge tracing. Thirdly, knowledge tracing can help improve educational gaming. Education gaming is a recent thing in the Internet age, which involves a game that is developed for a specific educational goal. Compared with traditional education methods, it uses games as a tool for education. To balance entertainment and education, educational gaming needs to adjust the difficulty level and concepts of the game based on the users' level so that they can learn the knowledge during the game. To accomplish this purpose, knowledge tracing can be used in tracking players' performance. And this can help developers to design a more effective and interesting educational game.

Figure 1 shows an example of context knowledge tracing, which contains four parts: (1) a set of questions with knowledge concepts and an interaction log of a student's corresponding practical behaviors, (2) the related elements extracted from the student's behaviors and questions, (3) the prediction model constructed for the elements, and (4) the predicted result for knowledge tracing. To be specific, a student has done several practices over a period of time. These practices are ordered by time and each question contains one or more knowledge concepts. Based on the mastery of these knowledge concepts, the student can give a right or wrong response to each question. We can collect these questions with knowledge concepts and students' corresponding responses as an interaction log to model the student's knowledge state. Besides this interaction log, students' mastery of knowledge is also affected by the Ebbinghaus Forgetting Curve [11]. Therefore, we can consider both interactions and the forgetting behavior to model students' performance and predict whether they can give a response to the next question correctly.

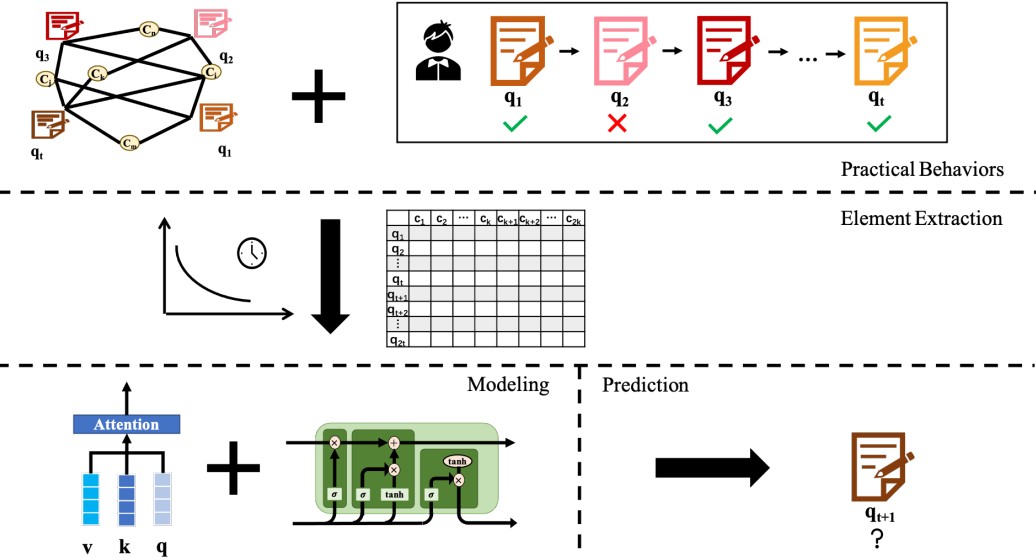

**Figure 1.** Overview of Context-based Knowledge Tracing.

Students' performances on a new question are based on their previous performances on the questions. To comprehensively capture the studying behaviors, we should combine the contexts of students' responses to historical questions with their knowledge masteries

and the knowledge concepts contained in each question. Recently, some models [12–14] have been designed for knowledge tracing using this information. Some of them analyze the interactions between students and questions for prediction. However, these models only consider practices and knowledge concepts on knowledge tracing, but ignore other factors that affect students' performances. In fact, there are multiple factors affecting the performance of the student, such as the student's memory and behaviors. In addition, their influence is a long-term effect on the study process. Therefore, knowledge tracing cannot be only modeled by the practices with corresponding knowledge concepts. The challenges are how to deal with the influence changes of these factors temporally and how to deal with the interaction influence of these factors. To involve multiple factors to predict students' performances accurately, we design the ContextKT model with more context factors on knowledge tracing. To address these challenges, we utilized the LSTM model to capture the context information of memory and forgetfulness, and the multi-head attention mechanism to capture the context information of students' behaviors. In addition, we also consider the personalized information in the questions for prediction, including the contained knowledge concepts and the similarity among questions.

In summary, the contributions of our work are given as follows:

(1) We propose a context-based approach for knowledge tracing. On the factors that affect students' performance, we define five contexts to build the prediction model and design the data structure and the models to describe or capture the context information.

(2) A QR-matrix is defined to represent the relationship among questions, knowledge concepts, and responses, which is used to describe the context information about questions and knowledge. An improved LSTM-based model is proposed to capture the context of students' memory and forgetfulness. A multi-head attention mechanism is designed to capture the context information of students' response behavior.

(3) A context-based prediction model ContextKT is established on top of the improved LSTM model and the multi-head attention mechanism. For optimization, the regularization item is designed to consider the question similarity and the graph Laplacian matrix is used to reduce the computation complexity.

(4) Our prediction model is evaluated on two real educational datasets. The experimental results show our model can achieve high performance on both datasets, and outperform the baseline methods.

The rest of the paper is organized as follows. We review the related works in Section 2. The problem of knowledge tracing is formulated in Section 3. Then we present our ContextKT in Section 4 and evaluate it in Section 5. Finally, we give the conclusion in Section 6.

## 2. Related Works

In this section, we introduce the works on knowledge tracing. Based on the techniques used in knowledge tracing methods, they can be divided into three categories: Bayesian Knowledge tracing (BKT) models, factor analysis-based models, and deep learning-based knowledge tracing models. In addition, we also introduce relevant studies which utilize context information in their works.

**Bayesian knowledge tracing models** usually utilize historical interaction records of students to model their knowledge states based on Hidden Markov Model, which is first provided by Corbett and Anderson [1]. There are also many works on improving the BKT-based model. Pardos and Heffernan [15] provide the Prior Per Student model, which can learn individualized parameters in a single step to accelerate the learning of global optimal fit parameters. To represent multiple skills within a single model, Käser et al. [16] designed a knowledge tracing model based on Dynamic Bayesian networks so that they can model prerequisite hierarchy and relationships among different skills in a learning domain. Getseva and Kumar [6] focus on saving time and problems in programming tutors with the help of BKT. For each question provided by tutors, they calculate slip and guess parameters in modeling based on an empirical approach to evaluate students' mastery states. Michael [5] improves BKT by adding the Elo component into the model, which

conduces to infuse students' problem-solving skills with BKT. In these BKT-based models, the knowledge states of students are treated as vectors for the Hidden Markov Model, which cannot represent the mastery of each individual knowledge concept, so these models cannot efficiently describe the students' mastery level of knowledge.

**Factor analysis-based models** take advantage of latent factors to predict the accuracy of students' records. For example, Cen et al. [17] improves a cognitive model by combining statistical model, human expertise, and A* search to model students' records. To trace individual students with their individual skills, Pavlik et al. [18] provided a performance factor analysis model. It is sensitive to the indicator of students learning performance by generating logit value so that this model can estimate the cost of each action in instructional engineering. Vie et al. [19] utilized factorization machines in the existing model, including AFM [17] and PFA [18], to deal with sparse students' data. Choffin et al. [7] involves both multiple skill tags and memory decay in the knowledge tracing model to extend DASH [20], which is a factor analysis-based model combining collaborative filtering and the psychological theory of memory. Ghost et al. [13] provided AKT, which embeds contexts by monotonic attention mechanism and utilizes the Rasch model to capture the difference among students on the same question. Unlike our model, AKT is an IRT-based model and we designed a deep learning-based model. In addition, the AKT input the embedding vector into the Rasch model for prediction, while we combine outputs of LSTM and multi-head attention mechanism to enhance accuracy. Compared with BKT-based models, IRT-based models involve the characters of students and questions, so these kinds of models are more interpretable than BKT-based models.

**Deep learning-based knowledge tracing models** utilize neural network techniques, such as Long and Short-Term Memory network (LSTM), Recurrent Neural Network (RNN), and Graph Neural Network (GNN) on the knowledge tracing tasks. They usually have a sequential structure, which can model the study process to retain the study trajectory information [21]. With the help of neural network techniques, deep learning-based models have a more powerful ability on cognitive processes. Deep Knowledge Tracing (DKT) [22] applied RNN to track students' knowledge states based on their learning history. Compared with BKT-based models, DKT can achieve high performance without human annotation. Ghosh et al. [4] provided SAKT, which uses a multi-head attention mechanism to model students' responses to historical questions. SAKT can study the relationship of the questions based on the attention mechanism so that it can support the sparsity of knowledge concepts. Zhang et al. [23] designed a memory-augmented model Dynamic Key-Value Memory Networks (DKVMN), which utilized a static matrix and a dynamic matrix to store knowledge concepts and students' mastery levels of corresponding concepts as key and value, respectively. Similar to DKVMN, He et al. [24] provided an Extensible Deep Knowledge Tracing model EDKT, which adds two plug-ins to provide features of the extended factors and extract learners' knowledge growth so that it can involve more learning factors to support various knowledge tracing tasks.

**Context involved methods.** Besides the AKT [13] model, there are also some methods [12,14,25,26] utilizing context information for knowledge tracing. Wong et al. [14] improve the AKT model and provide iAKT, which uses regularization strategies to learn students' performance distributions and incrementally trains AKT. Huang et al. [12] study knowledge tracing based on Bi-LSTM. They first split questions to generate corresponding embedding vectors, and then combine them with knowledge concepts and students' responses to input into Bi-LSTM for prediction. Krishnan et al. [26] utilize repeated time gap, sequence time gap, and past trial counts to describe context features and construct Bi-Interaction Deep Knowledge Tracing (BIDKT). Nagatani et al. [25] improve DKT model [22] and consider forgetting behavior, which is considered as context, in their model. However, not all these works have the same definition of contexts as us. The contexts in this paper focus on students' performances, response behaviors, the questions, and the knowledge concepts in the questions.

The factors that affect students' performances are complex and existing models usually consider one or few of them. To overcome this problem, in this paper, we propose ContextKT, which takes advantage of deep learning techniques to support more context factors on knowledge tracing.

## 3. Problem Formulation

In this section, we give the definitions of basic concepts and symbols used in the knowledge tracing.

**Knowledge tracing.** Knowledge tracing aims to predict a student's mastery of knowledge based on their previous behaviors in a time interval. Generally, we use the student's performance on the question at a specific time to represent their mastery of knowledge at that time, that is, whether they can give a response to the question correctly. In addition, the previous behaviors are represented by a historical practice log, including the questions with one or more knowledge concepts and corresponding responses. Then, we can formalize the definition of knowledge tracing.

Given a history log of student $i$ on time interval $[0, t]$, and his/her historical practice log contains a study sequence $S^i = \left\{ (q_1^i, r_1^i), (q_2^i, r_2^i), \ldots, (q_t^i, r_t^i) \right\}$ and information of questions such as the marked knowledge concepts, where $q_t^i$ is the question that a student $i$ answers at time $t$ and $r_t^i$ is the corresponding response at time $t$. knowledge tracing aims to predict his/her response $r_{t+1}^i$ on question $q_{t+1}^i$ based on his/her study responses during time interval $[0, t]$ to acquire the student $i$'s mastery of knowledge.

Based on this definition, the purpose of our knowledge tracing task is to predict the accuracy of the student's response to the next question $q_{t+1}^i$. Let $F$ denote students' performance model. Then the response accuracy $p(r_{t+1} \mid q_{t+1})$ can be calculated as $p(r_{t+1}^i \mid q_{t+1}^i) = F(\Theta; S; E)$, where $\Theta$ is the parameter of model, $S$ is the history response set of questions and $E$ is the set of relevance data on questions such as marked knowledge concepts and texts of questions.

The basic idea of the knowledge tracing model is to construct an effective model on students' responses to trace their knowledge state and predict their performance on the specific question at the next moment.

**Q-matrix** is a mapping matrix between questions and knowledge concepts. As a question may contain multiple knowledge concepts, Q-matrix represents the relationship between a question and its knowledge concepts. Consider a dataset with $N$ questions and $M$ knowledge concepts. A Q-matrix is constructed with $N$ rows and $M$ columns, in which each column represents a knowledge concept, and the row vector represents the concept information contained in a question. The Q-matrix is a binary matrix; considering an element at the $i$-th row $k$-th column, it represents whether the $i$-th question contains $j$-th knowledge concepts. If it contains these concepts, the element is 1. Otherwise, it is 0.

The mathematical notations used in this paper are summarized in Table 1.

**Table 1.** Notations.

| Parameters | Descriptions |
| :---: | :--- |
| $F$ | the predict model for students' performance |
| $E$ | the set of relevance data on questions |
| $S$ | the set of interaction sequences between students and questions |
| $\Theta$ | the parameter set of the model |
| $P$ | the set of questions |
| $s^i$ | the interaction sequence between student $i$ and questions |
| $q_t^i$ | the question that student $i$ responses at time $t$ |
| $r_t^i$ | the response from student $i$ on the question at time $t$ |
| $e_{\langle q_t, r_t \rangle}$ | the embedding of exercise interaction $\langle q_t, r_t \rangle$ |

**Table 1.** *Cont.*

| Parameters | Descriptions |
|:---:|:---|
| $C_{q_t}$ | the set of knowledge concepts corresponding to the question $q_t$ |
| $\hat{r}_t$ | the performance of the student on the questions $q_t$ at time $t$ |
| $\hat{x}_t$ | the context feature embedding of the student's interaction at time $t$ |
| $Qo$ | the question-concept relationship matrix |
| $QR$ | the extended Q-matrix by adding responses |
| $N$ | the total number of questions |
| $M$ | the total number of concepts |

## 4. ContextKT Framework

In this section, we provide a context-based knowledge tracing approach to predict students' performances. We first give the contexts for the model design. Then, we explain the details of each part in our model.

### 4.1. Contexts for Performance Prediction

Considering the definition of knowledge tracing, we can see the knowledge concepts that students have already mastered and their mastery levels are fixed at the moment. Therefore, to make the process of tracing have characteristics of human learning, we utilize Ausubel's meaningful learning theory [27] to design our model, whose mental mechanism is assimilation. Meaningful learning theory focuses on learning transfer, that is, the student's performance on the new question is based on their historical behaviors, including the interactions on the historical questions and their knowledge mastery. For the students' performances, students must answer the questions in time; otherwise, they would be judged to have answered incorrectly. In addition, we assume the students are honest in their practices. Based on these theories, we give five contexts used in the model design:

(1) $\mathscr{C}_1$ **(Current state).** Current knowledge state of a student is reflected by their performance on historical questions. It means if a student has a good grasp of a specific knowledge concept, they should achieve a great performance on the questions that contain this concept.

(2) $\mathscr{C}_2$ **(Memory and forgetfulness).** Based on the pedagogy theory, a student may learn some new knowledge and forget some old knowledge in the process of interaction.

(3) $\mathscr{C}_3$ **(Question-knowledge relevance).** The performance on the current question of a student is affected by his grasp of relevant knowledge concepts.

(4) $\mathscr{C}_4$ **(Behavior similarity).** Besides knowledge state, a student's performance on the current question is also affected by their historical responses.

(5) $\mathscr{C}_5$ **(Question similarity).** A student gives similar feedback on similar questions. That means, given two questions that contain the same knowledge concepts, if a student can answer one question correctly, they have a good chance of getting the other one right.

These contexts involve the students' states and performances and the relationships among questions, knowledge concepts, and responses. In these contexts, $\mathscr{C}_1$, $\mathscr{C}_3$, and $\mathscr{C}_5$ are explicit contexts that we can represent directly, while $\mathscr{C}_2$ and $\mathscr{C}_4$ are implicit and we need to deal with them in the process of knowledge tracing.

In this approach, the input is student's responses of previous questions, which is denoted by $\{\langle q_1, r_1 \rangle, \langle q_2, r_2 \rangle, \ldots, \langle q_t, r_t \rangle\}$, where $q_t$ is the question that the student answered at moment $t$, and $r_t$ is the response of this question $q_t$. To leverage this information, now we formalize the method design as follows:

For context $\mathscr{C}_1$, our model mainly considers the interactions in the time interval $[1, t-1]$ and the relevant information of question $q_t$ to predict a student's response $\hat{r}_t$ at time $t$. Therefore, we can predict the response as $p(\hat{r}_t) = F(\langle q_1, r_1 \rangle, \langle q_2, r_2 \rangle, \ldots, \langle q_t, r_t \rangle, q_t)$.

For context $\mathscr{C}_2$, we model the memory and forgetting based on the in-gate and forgetting-gate of LSTM.

For context $\mathscr{C}_3$, we utilize QR-matrix to describe knowledge concepts of questions. Then we embed this concept information and other relevant information of questions to generate the final embedding. The details will be discussed in Section 4.2.

For context $\mathscr{C}_4$, We will model the behavior similarity on a student's responses with the multi-head self-attention mechanism. The details will be discussed in Section 4.5.

For context $\mathscr{C}_5$, one solution is based on the regularization of question similarity, which will be discussed in Section 4.6. In addition, the other one is to involve question similarity through the QR-matrix of the common questions' embedding.

In summary, we first generate question embedding based on the basic information of each question. Meanwhile, we also generate interaction embedding based on the accuracy of the current response. Then, utilizing the multi-head attention mechanism, we can construct the context of the current question's interaction tuple. We upload this information into the LSTM-based sequence model to model students' exercise sequences and we can obtain the prediction of students' current knowledge state. To improve the performance of prediction, we utilize the attention mechanism-based model in the prediction phase and calculate model loss based on prediction results and actual labels. In addition, we design regularization of question similarity for common information among questions. The overall structure of our approach is shown in Figure 2.

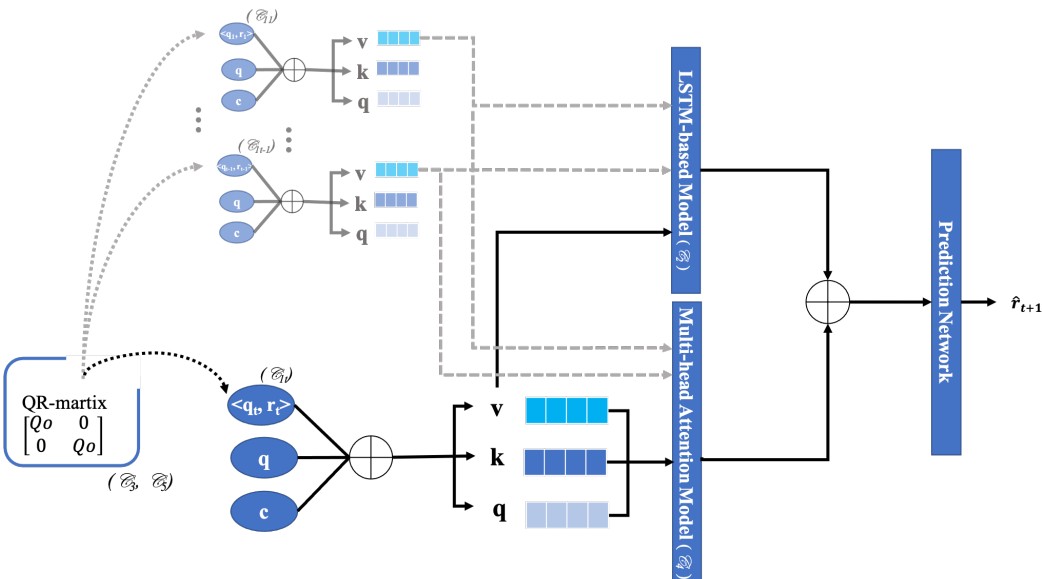

**Figure 2.** ContextKT Structure Diagram. We construct the model based on five contexts. To be specific, Context $\mathscr{C}_1$, $\mathscr{C}_3$ and $\mathscr{C}_5$ are related to the questions and students' corresponding performances. This information is treated as input in ContextKT. Context $\mathscr{C}_2$ and $\mathscr{C}_4$ are both process during study. So we use LSTM and multi-ahead attention to represent them.

### 4.2. Embedding Representation of Interaction by QR-Matrix

There are two strategies to represent the interactions between students and questions in the traditional knowledge tracing methods.

Suppose that the number of questions is $N$. At time $T$, the student is answering the question $q_t$, and his/her response is $r_t$. If this student can answer the question correctly, then we can use the ID of question $q_t$ to label this interaction. Otherwise, the interaction is labeled with $q_t + N$. In this way, we can model interaction by questions. Even though this method can involve personalized information of questions, it ignores the knowledge concepts contained in the questions.

Another strategy considers the knowledge concepts in the questions. Suppose the number of questions and the number of knowledge concepts are $N$ and $M$, respectively. It first transforms the question ID to the concept ID, and use the concept ID to represent the question. Then, for each interaction, it also utilizes the concept ID to represent it.

In this way, it considers that an interaction contains a knowledge concept and sparse data is avoided. However, this method lacks a proper way to deal with questions with multiple concepts. Moreover, it considers that the questions with the same concepts have a comparable level of difficulty, which ignores personalized information. To address these issues, we provide a concept-based embedding method on questions, which can consider both multiple knowledge concepts and personalized information of questions.

Since the original Q-matrix only can represent the relationship between questions and knowledge concepts, to distinguish the representations of questions with correct or wrong responses, we utilize students' responses to extend Q-matrix and obtain QR-matrix. Let $c_k$ denote the knowledge concepts contained in question $q_t$. If the student can answer $q_t$ correctly, we use $c_k$ to represent the knowledge concepts in the matrix. Otherwise, we represent it with $c_k + M$. It means that for the question with the correct response, it keeps the original length of the knowledge concept index. In addition, for the question with a wrong response, its index's length will increase $M$. Based on this structure, we can observe that the correlation between QR-matrix and Q-matrix is $QR = \begin{bmatrix} Qo & 0 \\ 0 & Qo \end{bmatrix}$, where $Qo$ is the original Q-matrix and 0-matrix is the matrix that has the same size with $Qo$ and each element is 0. In the QR-matrix, we utilize interaction tuple $\langle q_t, r_t \rangle$ to index knowledge concepts. For the questions with wrong responses, we use $q_t + N$ to index knowledge concepts contained in it and we can obtain concept index from $c_k$ to $c_k + M$. By utilizing $q_t$ to index the questions with correct responses, we can obtain the original concepts index $c_k$. After obtaining all the concept indexes of the corresponding interaction, we can use them to embed knowledge concepts. We can obtain the initial question embedding through the sum of concept embeddings. By summing personalized embedding of questions, we can obtain the final embedding, that is $e_{\langle q_t, r_t \rangle} = e_{q_t} + \sum_{c_k \in C_{\langle q_t, r_t \rangle}} e_{c_k}$, where $C_{\langle q_t, r_t \rangle}$ denotes the knowledge concepts contain in the question $q_t$ whose response is $r_t$ at time $t$, $e_{c_k}$ denotes the embedding of concept $c_k$, $e_{q_t}$ denotes the personalized embedding of question $q_t$, and $e_{\langle q_t, r_t \rangle}$ is the final embedding of question interaction $\langle q_t, r_t \rangle$.

*4.3. LSTM Based Modeling for Memory and Forgetfulness*

We denote the practice sequence of student $i$ with Function (1). Then, based on the embedding method of students' question interaction tuple, we can generate interaction embedding as shown in Function (2).

$$S^i = \{ \langle q_1^i, r_1^i \rangle, \left( q_2^i, r_2^i \right), \ldots, \langle q_t^i, r_t^i \rangle, \ldots, \langle q_T^i, r_T^i \rangle \}. \tag{1}$$

$$S_e^i = \{ e_{\langle q_1, r_1 \rangle}, e_{\langle q_2, r_2 \rangle}, \ldots, e_{\langle q_t, r_t \rangle}, \ldots, e_{\langle q_T, r_T \rangle} \}. \tag{2}$$

where $i$ denotes the student's id. For simplicity, in the reminder of paper, we will omit $i$ in the following functions.

As we discussed before, the student's performance on the current question is affected by forgetfulness. Therefore, we utilize the LSTM model to context $\mathscr{C}_2$. After generating the embedding representation of the interaction sequence between student and questions, we can generate the modeling process of LSTM for the student's practice log with the following functions:

$$f_t = \sigma(W_{ef} e_{\langle q_t, r_t \rangle} + W_{hf} h_{t-1} + b_f). \tag{3}$$

$$i_t = \sigma(W_{ei} e_{\langle q_t, r_t \rangle} + W_{hi} h_{t-1} + b_i). \tag{4}$$

$$o_t = \sigma(W_{eo} e_{\langle q_t, r_t \rangle} + W_{ho} h_{t-1} + b_o). \tag{5}$$

$$\hat{c}_t = f_t \cdot \hat{c}_{t-1} + i_t \cdot \tanh(W_{e\hat{c}} e_{\langle q_t, r_t \rangle} + W_{h\hat{c}} h_{t-1} + b_{\hat{c}}). \tag{6}$$

where $W_{e*} \in R^{d_h \times d_e}$, $W_{h*} \in R^{d_h \times d_h}$, and $b_* \in R^{d_h}$ are the adjustable parameters in LSTM, and $h_t$ and $\hat{c}_t$ are the hidden state and the cell state of LSTM at time $t$, respectively. Based on

the modeling of the student's historical practice log, we can generate the advanced feature for interaction sequence with the following function:

$$S_h = \{h_1, h_2, \ldots, h_t, \ldots, h_T\}. \tag{7}$$

### 4.4. Positional Embedding

To enable the structure, such as context, to take advantage of the order information of the sequence, we add relative or absolute position information to each node embedding in the sequence. The positional embedding has the same dimensions as the embedding. Usually, there are two types of positional embeddings, learnable and fixed. In this paper, we employ sine and cosine functions with different frequencies to construct positional embeddings.

$$
\begin{aligned}
PE_{(pos,2i)} &= \sin\left(pos/10{,}000^{2i/d_{\mathrm{model}}}\right) \\
PE_{(pos,2i+1)} &= \cos\left(pos/10{,}000^{2i/d_{\mathrm{model}}}\right)
\end{aligned}
\tag{8}
$$

where *pos* is the position and *i* is the dimension. This type of positional embedding parameter can make it easier for the model to learn attention weights based on relative positions. Because of this positional embedding, $PE_{pos+k}$ can be expressed as a linear function of $PE_{pos}$ for a fixed offset $k$.

### 4.5. Multi-Head Attention Mechanism-Based Modeling for Behaviors

Representation of learning of context is based on this assumption: The student's response to the current question is related to their recent responses to questions. These responses can reflect both students' knowledge state and their psychological state on the current question. For the psychological state, we generate it from the law of effect [28], that the results the learner achieves will affect their positivity. That means, suppose there are two students, A and B have the same knowledge states on time$t$. However, the accuracy rate of A's responses increased in time interval $[0, t-1]$, while B's is on the contrary. Due to a series of correct/wrong responses, A may have a more positive attitude while B may be negative. Therefore, these two students may have different responses to the same question at time $t$.

Our representation learning model of context embedding is designed based on the self-attention mechanism. Even though LSTM can model the student's memory and forgetfulness based on in-gate and forgetting-gate, self-attention achieves higher performance on mining correlations among interactions within the long interval. Therefore, to comprehensively describe the interactions between students and questions, we model the context of interaction sequences with a multi-head attention mechanism. For the embedding of interaction, we contact current question and knowledge concepts as the query vector, which is also treated as a key vector. The value vector is the concat of interaction embedding, current question, and knowledge concepts. Then we calculate the dot products between the current query vector and each past key vector and use the softmax function to generate the weights of attention. For each weight, we multiply it with the corresponding value to generate the relevance vectors. Finally, we calculate the weighted sum of the relevant vector and the embedding of each historical interaction to generate the context feature embedding of the current interaction. Based on this process, let $Q \in R^{T \times d_q}$, $K \in R^{T \times d_k}$, and $V \in R^{T \times d_v}$ denote query matrix, key matrix, and value matrix, respectively. In addition, let $d_q$, $d_k$, $d_v$ denote query vector, key vector and value vector, respectively. In addition, we make $d_q$ equal to $d_k$ to calculate the current relevance between the query vector and key vector. In addition, let $T$ denote the length of the interaction sequence. Therefore, we can calculate the single head attention mechanism as $Attention(Q, K, V) = softmaxe\left(QK^T / \sqrt{d}\right)V$.

In addition, different from other areas, when utilizing the attention mechanism in the knowledge tracing area to calculate current embedding, we cannot consider future

interactions. That means, when calculating context embedding on time $t$, we only can use the embedding in time interval $[0, t-1]$ and the question $q_t$ at time $t$, but not the embedding of time interval $[t+1, T]$ or the response $r_t$ for the question $q_t$. Therefore, we utilize masks to hide the influence of the past on current interaction, that is

$$MultiHead(Q, K, V) = concat(head_1, \ldots, head_i, \ldots, head_H)W^o. \tag{9}$$

$$head_i = Attention(QW_i^Q, KW_i^K, VW_i^V). \tag{10}$$

As the single-head attention mechanism is too simplistic, which limits its expression of the feature vector, we utilize the multi-head attention mechanism in our approach. In the multi-head attention mechanism, every single one has independent weight. Therefore, it can describe the correlation between current interactions and historical interactions from multi-perspective views with its powerful expression. Figure 3 shows the computation process of the multi-head attention mechanism. The multi-head attention mechanism is utilized to model behavior similarity $\mathscr{C}_4$, which is an implicit context involved in the context feature embedding of a student's current interaction. In addition, in every single head, it involved different implicit factors of $\mathscr{C}_4$ and we use $\mathscr{C}_{4i}$ to denote each output.

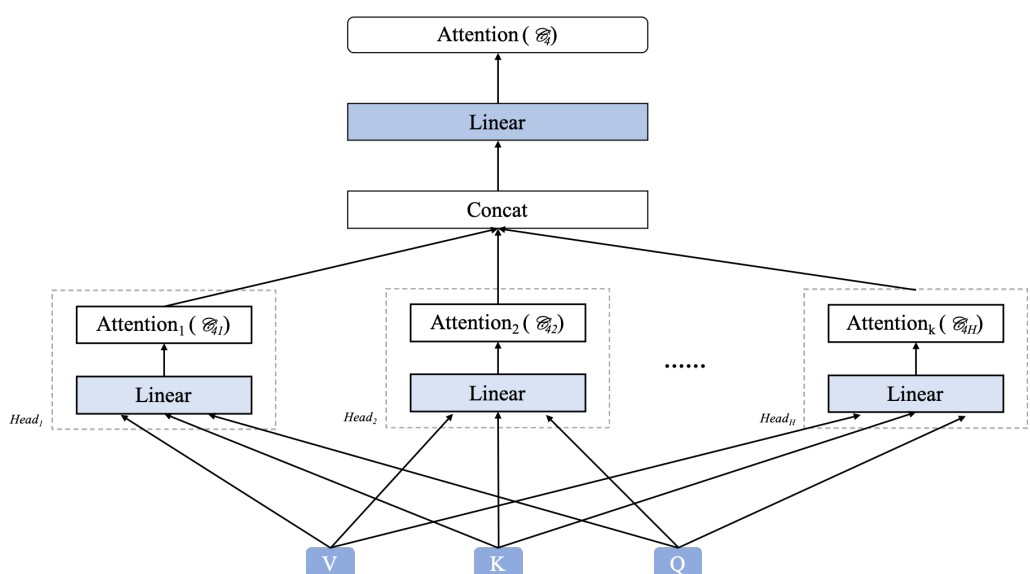

**Figure 3.** Multi-head Attention Mechanism for Capturing Context $\mathscr{C}_4$.

In the multi-head attention mechanism, it maps original input vectors to $H$ queries, keys and values with different linear layers. We use Functions (9) and (10) to describe it, where $H$ is the number of heads, $head_i$ is the embedding representation of a single head, and $Q = K = V = e_{\langle q_t, r_t \rangle}$ is the input of multi-head attention mechanism. With these functions, we can use a single head mechanism to calculate vectors after mapping, then connect these outputs from different attention mechanisms to generate a more powerful feature vector.

Let $\hat{x}_t$ denote context feature embedding of the student's interaction at time $t$. Utilizing Functions (9) and (10), we can generate an embedding representation of context information at each time in time interval $[0, T]$ as shown in Function (11):

$$S_{\hat{x}} = \{\hat{x}_1, \hat{x}_2, \ldots, \hat{x}_t, \ldots, \hat{x}_T\}. \tag{11}$$

In this paper, we obtain feature embedding at each time with Function (12). In addition, for the feature embedding of the interaction sequence between students and questions, we obtain with Function (13).

$$z_t = h_t + \hat{x}_t. \tag{12}$$

$$S_z = \{z_1, z_2, \ldots, z_t, \ldots z_T\}. \tag{13}$$

### 4.6. Regularization of Question Similarity

To construct this part, we suppose that the student has similar performances when responses similar questions. The more similar questions are, the more similar the performances are. In the meantime, after analyzing the performance of our method with different datasets, we observe that if the number of questions is too much and each question has been answered too little, the performance of the method will deteriorate. To overcome this problem, we formalize the similarity of questions to improve the method's performance, which can calculate the bias of accuracy among the questions with the same knowledge concepts. Function (14) shows the process of regularization:

$$R(y) = \sum_{i \in P} \sum_{j \in P} \mathbb{I}(i, j)(y_i - y_j)^2. \tag{14}$$

where vector $y \in R_N$ is the probability that our method predicts that the student will answer the questions from the dataset correctly, $P$ are the set of questions in the dataset. $\mathbb{I}(i, j)$ is the indicator vector. It turns to 1 if questions $i$ and $j$ contain the same knowledge concepts. Otherwise, it turns to 0.

As the complexity of Function (14) is $\mathcal{O}(|P|^2)$ with the double sum, we optimize the regularization item with graph theory to speed up the calculation. We first construct a graph $G$ with $P$ vertices, and each vertex in $G$ represents a question in the dataset. If two questions share the same knowledge concepts, then the corresponding vertices will have an edge with weight 1. Let diagonal matrix $D$ become the degree matrix of a graph; it can be calculated as shown in Function (15), where $w_{ij}$ is the similarity between vertex $i$ and $j$, and $CV_i$ is the set of vertices connecting with vertex $i$. For the weight of edges, we use the adjacent matrix $A$ to represent. Then, we can define the Laplacian matrix as shown in Function (16).

$$d_{ii} = \sum_{j \in CV_i} w_{ij}. \tag{15}$$

$$L = D - A. \tag{16}$$

For any vector $v$, let $v_i$ denote the value of vertex $i$ in the graph. We can generate inference based on the Laplacian matrix as Function (17).

$$v^T L v = \sum_{i, j} w_{ij}(v_i - v_j)^2. \tag{17}$$

After combining this inference into regularization Function (14), we can transform it into matrix multiplication as Function (18).

$$R(y) = y^T L. \tag{18}$$

where $R(y)$ is regularized value, $i$ and $j$ are the questions in the question set $P$. Based on matrix multiplication, we can speed up the calculation of regularization terms, especially in the experimental environment using GPU.

### 4.7. Prediction for Student's Performance

Usually, we can represent the prediction model in the knowledge tracing method with Function (19):

$$p(\hat{r}_{t+1}) = predict(z_t). \tag{19}$$

where $z_t$ is the feature representation from historical responses and $\hat{r}_{t+1}$ is the predicted results of the student's response on the question at time $t + 1$. Based on this function, we utilize the current feature in the interaction sequence of students to predict their performance in the next moment.

In our method, we consider the interaction sequence between the student and questions has Markov characteristics. That means the student's performance at the next moment is only related to that in the current moment, but not time $t - 1$ or before. However, the prediction strategy based on Markov characteristics is not compatible with the LSTM-based knowledge tracing model. Especially when the question sequence is too long, it always loses some information in the LSTM. Therefore, we utilize the attention mechanism-based prediction model in our method to enhance the influence of the interaction in the historical question interactions, which has a strong correlation with the current question. The prediction strategy is shown as Functions (20) and (21).

$$h_{att} = \sum_{j=1}^{t} \alpha_j h_j. \tag{20}$$

$$\alpha_j = \cos\left(e_{(q_t,r_t)}, e_{(q_j,r_j)}\right) = \frac{e_{(q_t,r_t)} \cdot e_{(q_j,r_j)}}{\mid e_{(q_t,r_t)} \mid \cdot \mid e_{(q_j,r_j)} \mid}. \tag{21}$$

where $\alpha_j$ is the similarity between the question embedding at time $t + 1$ and $j$. It can be used to measure the influence of question interaction at time $j$ to the question on current time $t$. Then, we combine the models above to predict the student's performance. To keep the accuracy and convergence quickly, we use the residual network in our model. Therefore, our prediction can be calculated based on Functions (22) and (23):

$$\hat{y}_{t+1} = ReLU(W_1[h_t, h_{att}] + b_1). \tag{22}$$

$$\hat{r}_{t+1} = \sigma(W_2\hat{y}_{t+1} + b_2). \tag{23}$$

where $\hat{r}_{t+1}$ is the student's performance at time $t + 1$. In addition, the feature embedding of the student's interaction at each moment will be the input into two fully connected layers with ReLU and sigmoid, respectively.

### 4.8. Loss Function

Our method utilizes Back Propagation (BP) to train the neural network. As student performance prediction is usually considered as the classification task in the knowledge tracing methods, which utilizes the cross-entropy loss function to calculate the loss. Our loss function is designed as shown in Function (24).

$$\mathcal{L} = \sum_{t} \ell(\hat{r}_t^{\mathrm{T}} \delta(q_t), r_t). \tag{24}$$

where $l$ is the cross entropy loss function, $\delta(q_t)$ is the one-hot code of question $q_t$ that student answered at time $t$, $r_t$ is correct label of question answered at time $t$, and $\hat{r}_t$ is the prediction result of students' performance at time $t$, which contains results on each question. Therefore, $\hat{r}_t^{\mathrm{T}} \delta(q_t)$ can be used to select the prediction result of student's performance that gives a response to the question at time $t$.

As we assume above, a student has similar performances on similar questions. We can predict the regularization items of results $\hat{r}_t$ with the regularization strategy in Section 4.6, that utilizes the predicted result $\hat{r}_t$ as the input of regularization item and calculate $R(\hat{r}_t)$ according to whether the questions contain common knowledge concepts and the differences among questions' mastery level predicted by the model. The regularization item is:

$$R(\hat{r}_t) = \sum_{i \in P} \sum_{j \in P} \mathbb{I}(i,j)\left(\hat{r}_i - \hat{r}_j\right)^2 = \hat{r}_t^{\mathrm{T}} L \hat{r}_t. \tag{25}$$

Furthermore, we utilize hyper-parameter $\lambda$ to control the weight of regularization in the loss function. Finally, our loss function is:

$$\mathcal{L} = \sum_t \ell(\hat{r}_t^{\mathrm{T}} \delta(q_t), r_t) + \lambda R(\hat{r}_t). \tag{26}$$

## 5. Experiments

In this section, we design multiple experiments to verify the effectiveness of our approach on two real educational datasets. We also compare our ContextKT with existing methods to evaluate its performance.

### 5.1. Experimental Setup

#### 5.1.1. Datasets

We use two real educational datasets, HDU and Algebra08, to evaluate our approach. HDU is a real dataset that collected the real interaction data between students and questions in the HDU online testing system. This dataset set contains data from June 2018 to September 2018. In addition, all the questions can be repeating practiced. Algebra08 is from the EDM challenge in the KDD Cup 2010, which collected interaction responses of quizzes from September 2018 to November 2018.

Moreover, in HDU, students are free to choose questions to practice so, their response accuracies are not high. Besides, the number of attempts on each question is sparse. Meanwhile, it has an obvious tendency on the questions that students try in Algebra08. This is because these data are collected from quizzes and the number of attempts on some questions are obviously higher than others.

To optimize the dataset, we filter the students whose accuracies are less than 10% or the number of whose interactions are less than 30. In addition, the questions answered less than 30 are also pruned. More details about these two datasets are shown in Tables 2 and 3.

**Table 2.** Dataset statistics of HDU.

| Categories | Size |
| --- | --- |
| students | 9859 |
| questions | 2101 |
| interaction responses between students and questions | 1,042,661 |
| knowledge concepts | 193 |
| avg interactions of students | 105 |
| avg interactions of questions | 496 |

**Table 3.** Dataset statistics of Algebra08.

| Categories | Size |
| --- | --- |
| students | 934 |
| questions | 178 |
| interaction responses between students and questions | 602,076 |
| knowledge concepts | 179 |
| avg interactions of students | 644 |
| avg interactions of questions | 1592 |

#### 5.1.2. Baseline Methods

To evaluate our context-based knowledge tracing method's performance, we select eight classic models for the comparative experiments.

- Item Response Theory(IRT) [29] is a classical cognitive diagnostic model which discovers students' cognitive level through logistic-like function.

- Multidimensional Item Response Theory (MIRT) [30] improves the IRT model. It extends scalars in IRT, which represents students' abilities and questions' difficulty with vectors.
- Additive Factor (AFM) [17] is a factor analysis model which mainly considers the number of attempts.
- Performance Factor Analysis (PFA) [18] is also a factor analysis model. Different from AFM, it considers the number of both correct and wrong attempts by students.
- Knowledge Tracing Machines (KTM) [19] integrates IRT, AFM, and PFA model. It can specialize each of them through parameters' setup.
- DAS3H [7] is a factor analysis model. It considers multiple factors, including memory loss and multi-knowledge concept labels.
- DKT [22] is the first model that utilizes deep learning techniques in the knowledge tracing area. It can track the change in students' knowledge mastery states over time through the LSTM model.
- SAKT [4] model is constructed based on the self-attention mechanism, and uses it to model students' responses and predict their feature performances.
- DKVMN [23] model is constructed based on a key-value memory network. It can obtain students' mastery level of each potential knowledge concept through the relationship among these potential concepts.

### 5.1.3. Evaluation Indices

To evaluate our ContextKT model's performance, we implement it on regression and classification tasks, respectively. For regression, we use Mean Absolute Error (MAE) and Root Mean Square Error (RMSE) to measure the gap between predicted and actual results. MAE and RMSE are calculated as shown in Functions (27) and (28).

$$MAE = \frac{1}{m} \sum_{i=1}^{m} | r_i - \hat{r}_t | \tag{27}$$

$$RMSE = \sqrt{\frac{1}{m} \sum_{i=1}^{m} (r_i - \hat{r}_t)^2} \tag{28}$$

where $m$ is the number of questions that students have answered, $r_i$ is the real score on question $i$, and $\hat{r}_t$ is the predicted score on question $i$. Therefore, the lower the MAE and RMSE are, the higher the performance our method achieves.

For classification, we can use our method of binary classification. We treat the case with a student's score of 1 as positive, and the case with a score of 0 as negative. To evaluate the performance, we use Area Under Curve (AUC), Accuracy (ACC), and Precision (PRE) as evaluate indices. AUC is defined by the area consisting of the ROC curve and the coordinate axis. ACC and PRE are calculated as shown in the following functions:

$$ACC = \frac{TP + TN}{n} \tag{29}$$

$$PRE = \frac{TP}{TP + FP} \tag{30}$$

where $TP$, $TN$, and $FP$ are the number of true positive, true negative, and false positive results, respectively. In addition, $n$ is the total number of results. Based on these equations, we can see the higher these three indices are, the better the performance our method has.

### 5.1.4. Evaluation Parameters

Parameters in our context-based knowledge tracing method are set as follows: In our ContextKT method, the embedding demission of interaction tuple between students and questions is 80. In our LSTM model, hidden state demission is 80, the size of dropout is 0.01, and the size of L2 regularization is 0.01. In the multi-head attention mechanism model,

the number of the head on Algebra08 and HDU are 3 and 5, respectively. In the process of training, we set the training period to 200 for convergence. For the Adam optimizer parameter, the learning rate is 0.001 and the learning rate attenuation is $1e^{-7}$.

### 5.2. Evaluating Head Number on ContextKT

In this subsection, we evaluate the influence of head number in the multi-head attention mechanism that we use in our context-based method. Figure 4 shows the curve of AUC changed with a head number on Algebra08. In addition, Figure 4a,b are the results of the training dataset and test dataset, respectively.

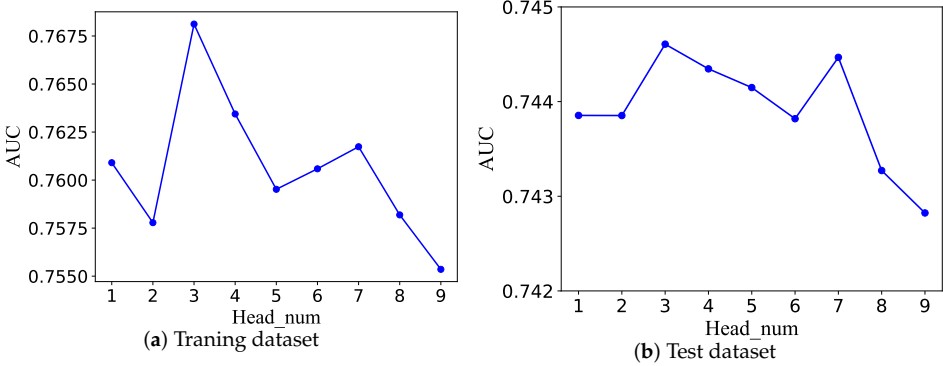

**Figure 4.** Influence of Head Number on ContextKT's Performance.

From the experimental results, we can see AUC on both two dataset increases first and then decrease with the head number increasing, and the optimal number of the head is 3. On HDU dataset, we find the same observation, and the optimal number is 5. Therefore, in the rest of the experiments, we will set the head number of our models 3 and 5 on Algebra08 and HDU, respectively.

### 5.3. Ablation Study

To evaluate the performance of each component in ContextKT, in this subsection, we conduct several ablation experiments. As the model consists of two components: LSTM and multi-head attention mechanism, we implement two variations of ContextKT, in which LA refers to the model that removes LSTM and its attention mechanism and CT refers to the model that removes the multi-head attention mechanism. The results are shown in Table 4. And the optimal value of the evaluation index is highlighted in bold.

**Table 4.** Ablation Study.

| Model | HDU | | Algebra08 | |
|---|---|---|---|---|
| | ACC | AUC | ACC | AUC |
| ContextKT | **0.685405** | 0.707121 | **0.813716** | **0.751075** |
| LA | 0.682909 | **0.709747** | 0.806055 | 0.750294 |
| CT | 0.673326 | 0.693555 | 0.800743 | 0.731580 |

From the results, we can see the performance of ContextKT is mainly inflected by LSTM. That means the study is a process of remembering and forgetting, mainly affecting students' mastery of knowledge. In addition, it is also other factors that affect knowledge mastery. We can see that the variation of LA achieved the worst performance in the experiment but it still can help our model improve its performance. This verifies the behavior of students and is a useful part of knowledge tracing.

*5.4. Comparison with Existing Methods*

In this subsection, we compared ContextKT with baseline methods. In the experiments, we evaluated the performance of ContextKT on all five evaluate indices: AUC, ACC, PRE, MAE, and RMSE. The results are shown in Tables 5 and 6.

**Table 5.** Performance Comparison of Methods on HDU Dataset.

| Model | AUC | ACC | PRE | MAE | RMSE |
|---|---|---|---|---|---|
| ContextKT | **0.707121** | **0.685405** | **0.657779** | 0.421046 | **0.459389** |
| DKT | 0.688482 | 0.662843 | 0.631488 | 0.420853 | 0.464716 |
| DKVMN | 0.693316 | 0.668589 | 0.634472 | **0.415228** | 0.463205 |
| SAKT | 0.683009 | 0.661207 | 0.633761 | 0.426612 | 0.463929 |
| KTM | 0.67722 | 0.657631 | 0.612784 | 0.430281 | 0.466044 |
| AFM | 0.52354 | 0.548486 | 0.421101 | 0.472343 | 0.554354 |
| DAS3H | 0.68344 | 0.659335 | 0.608092 | 0.426824 | 0.463968 |
| IRT | 0.642549 | 0.641442 | 0.600058 | 0.441641 | 0.473463 |
| MIRT | 0.642874 | 0.641358 | 0.590304 | 0.451555 | 0.473396 |
| PFA | 0.563187 | 0.570269 | 0.456779 | 0.449813 | 0.559884 |

**Table 6.** Performance Comparison of Methods on Algebra08 Dataset.

| Model | AUC | ACC | PRE | MAE | RMSE |
|---|---|---|---|---|---|
| ContextKT | **0.751075** | **0.813716** | **0.825841** | **0.271567** | **0.370025** |
| DKT | 0.738981 | 0.812905 | 0.824963 | 0.27309 | 0.371288 |
| DKVMN | 0.739936 | 0.810656 | 0.814686 | 0.277212 | 0.371782 |
| SAKT | 0.721765 | 0.808951 | 0.814686 | 0.276537 | 0.375799 |
| KTM | 0.708104 | 0.808217 | 0.812423 | 0.286546 | 0.378445 |
| AFM | 0.548704 | 0.732023 | 0.809430 | 0.330467 | 0.465264 |
| DAS3H | 0.710522 | 0.808474 | 0.812075 | 0.285311 | 0.377648 |
| IRT | 0.65186 | 0.805118 | 0.807628 | 0.301519 | 0.386675 |
| MIRT | 0.650422 | 0.804997 | 0.806328 | 0.311088 | 0.387262 |
| PFA | 0.590877 | 0.741067 | 0.816862 | 0.309607 | 0.452905 |

The results on HDU are shown in Table 5. In the classification task, ContextKT outperformed all the baseline methods. In addition, in the regression task, the ContextKT model achieved the highest performance of other methods on all evaluation indices except MAE. Compared with MAE, RMSE can better reflect whether there is a serious deviation value in the process of prediction. Therefore, ContextKT will not make serious mistakes in the prediction of students' performance. We can see the deep-learning-based methods outperform the IRT-based methods. This is because the ability of a student is usually modeled as a constant in the IRT-based methods., which is handled dynamically in the deep learning-based models. Such as, in the IRT, they treat students' abilities as a one-dimensional vector and they do not consider the connection between questions and knowledge concepts. Therefore, they do not achieve a good performance in the experiments. DKT into the study records of a student into RNN to obtain their knowledge states, which can effectively capture the current ability of the student. Thus DKT outperforms IRT in the experiments. Our method utilizes deep learning components to capture the multiple factors in the study process of students, including both the behaviors and changing of memory. Thus, ContextKT achieves the best performance in the evaluations.

Table 6 shows the comparison results on Algebra08, which is similar to the ones on HDU. To be specific, we can see ContextKT achieved the highest performance on all evaluation indices than other methods in both classification and regression tasks. Then DKT, DKVMN and SAKT are better than others. This is because deep learning-based models, including ContextKT, DKT, DKVMN, and SAKT, have better performances than models based on traditional machine learning. For example, the AUC of deep learning-based

models usually are higher than 0.7, while traditional models, such as IRT, are around 0.6. In addition, it shows the superior performance of deep learning techniques in the knowledge tracing area. For the IRT-based models, PFA and AFM are better than IRT as they improve IRT by involving students' practice records and the number of correct/wrong attempts. KTM is better than these three methods since it integrates all of them to enhance the ability of IRT. For the deep learning-based model, DKVMN process both fixed knowledge concepts and dynamic knowledge level, while DKT uses the RNN model to trace students' dynamic knowledge state so that they achieve a good performance in the evaluation. SAKT model the previous study sequence and consider the relationship between questions for knowledge tracing. However, SAKT's performance is slightly lower than DKT and DKVMN, maybe because our datasets are not the questions that rely on memory, which means the experiments depend on much longer records. Our ContextKT models both students' memory and forgetness in the study process with the LSTM-based model and considers the relationship between both questions and students' behaviors. Therefore we achieve the best performance in the evaluations.

## 6. Conclusions

In this paper, we design a context-based knowledge tracing method, ContextKT, to predict students' performances based on LSTM effectively. To capture more context information, we design QR-matrix to represent the relationship among students' responses, questions, and knowledge concepts. Our model considers the behavior similarity context by utilizing a multi-head attention mechanism on students' responses to the interaction sequences between students and questions. As the performance on questions is related to the students' knowledge mastery, we consider the knowledge concepts in the questions for prediction. In addition, we also consider question similarity when modeling the interaction sequence of questions. The experimental results show that ContextKT can effectively predict students' performances. Our model achieves a high performance on the real educational datasets, and outperforms the existing models. In the future, we will involve more viewpoints of psychology to study further the influence of students' mental activities and personality traits on knowledge tracing. Besides, we will consider more goals such as the speed and depth of knowledge's mastery in the knowledge tracing.

**Author Contributions:** Data curation, F.L.; Formal analysis, H.L.; Funding acquisition, M.Y. and T.Z.; Methodology, M.Y. and F.L.; Software, F.L.; Supervision, T.Z. and G.Y.; Writing—original draft, M.Y. and F.L.; writing—review and editing, M.Y. All authors have read and agreed to the published version of the manuscript.

**Funding:** This research was funded by the National Natural Science Foundation of China (No. U1811261, No. 61902055, No. 62137001), and Fundamental Research Funds for the Central Universities (No. N2117001).

**Institutional Review Board Statement:** Not applicable.

**Informed Consent Statement:** Not applicable.

**Data Availability Statement:** The datasets can be obtained from https://acm.hdu.edu.cn (HDU), and https://pslcdatashop.web.cmu.edu/KDDCup/downloads.jsp (algebra08), (accessed on 8 October 2020).

**Acknowledgments:** We really appreciate the valuable comments from the anonymous reviewers, which makes the paper more comprehensive.

**Conflicts of Interest:** The funders had no role in the design of the study; in the collection, analyses, or interpretation of data; in the writing of the manuscript; or in the decision to publish the results

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
