# Peer review of "ContextKT: A Context-Based Method for Knowledge Tracing"

_applsci, doi:10.3390/app12178822_

Round 1

Reviewer 1 Report

Article focus is on intelligent knowledge tracing to predict the students performance. There is no clarity in the problem statements, if authors includes the research challenges in students’ knowledge masteries or research questions taken for this study will add better understanding on the research work.

In related work section, each of the existing works result analysis is missing. Limitation of the existing works are not discussed. Author must include the research gaps from knowledge tracing approaches and how it is overcome by proposed method-LSTM. 

In ContextKT framework, few mathematical formulae, the parameters are not well defined, need to double check.

Multiple experiments were conducted to verify the effectiveness of the approach, there is clear analysis on accuracy of the proposed method and previous methods. 

Author Response

Dear Reviewer:

Really appreciate you for the decision of allowing a resubmission of our manuscript and really appreciate the valuable comments and suggestions from you and reviewers. We carefully revised the manuscript.

We are uploading (a) our point-by-point response to the comments (below) (response to reviewers), (b) an updated manuscript with blue highlighting indicating changes, and (c) a clean updated manuscript without highlights (PDF main document).

  1. Article focus is on intelligent knowledge tracing to predict the students performance. There is no clarity in the problem statements, if authors includes the research challenges in students’ knowledge masteries or research questions taken for this study will add better understanding on the research work.

Author response:  For clear description about the problem statements, we improve Sec 1 by rewriting the research challenges. And we revise the definition of knowledge tracing in Sec 3 by describing its formal definition in more detail.

  1. In related work section, each of the existing works result analysis is missing. Limitation of the existing works are not discussed. Author must include the research gaps from knowledge tracing approaches and how it is overcome by proposed method-LSTM. 

Author response: We add the discussion of related works to analyze their advantages and disadvantages in Sec 2. And we also introduce the research gaps and the differences between the related works and our model in Sec 2.

  1. In ContextKT framework, few mathematical formulae, the parameters are not well defined, need to double check.

Author response: We are sorry for the errors in our expression. We update the notation list to add some missing notation definitions and clean up ambiguities in Table 1.

  1. Multiple experiments were conducted to verify the effectiveness of the approach, there is clear analysis on accuracy of the proposed method and previous methods. 

Author response:  We add more analyses of the experiment results and explain the reason of differences in the performance in Sec 5. We also give the future works in Sec.6.

Reviewer 2 Report

The paper represents interesting and consistent study, with clear methodology explained in detail, with a rigourous results' presentation and the conclusions based on the results. This research is worth publishing, and today the implementation of neural network tools is relevant, especially for the e-learning platforms. The automated analysis helps to improve the path of the knowledge mastering and the procedures of assessment of the education quality and individual outcomes.

The positive impression of the material could be better, if the discussion of the results would present a deeper analysis of the outcome of the study.

The limitations of the research should also be mentioned, such as answers on several questions regarding the methodology (or these remarks could be enumerated as limitations of the study):

- do the authors measure the impact of repeating? If many questions concerns an issue (a concept), what is the influence of number of iterations to increase the grasp and the mastery of knowledge?

- do the authors classify the students according (1) the speed of grasping and depth of mastery of knowledge, and (2) their psychic type: the intro/extra-version, neurotism, et other features of personality? The "psychological state" described in section 4.5 is not clear.

- does the T and t differ from student to student, from level to level, from discipline to another? Is T and t limited, i.e., one minute for a question? Is the time to answer a question is taken into accout, e.g., if a student is asnwering too long, what happens, the answer is marked as wrong? Or not, the period can be infinite?

- do the researchers take into account a potential fraud?

These remarks could improve the content, giving more correct understanding of the further options to implement the outcome.

The last remark concerns many typos to correct, such as:

line 6 - "thier" - may be, their?

58 - "exmaple" - example?

75 - "inforamtion" - information?

line 81 - "affact students’ performane" - affect students’ performance?

82 - "and designd" - design?

85 - "to describes" - to describe?

87 - "fogetness" - "forgetness"

91 - "desinged" - designed?

93 - edcuational

100 - "traing" - tracing?

107 - "... and Andercon [14]" - Anderson

199 - "Meningful" - meaningful?

202 - "desgin" - design

205 - "he/her" - he/she

219 - "explicit contexts that we can represent them directly" - that we can represent directly - or - and we can represent them directly

233 - "similariy" - similarity

242 - "exercis" - exercise

294 - "influened"

361 - "similarr"

455-456 - " It can specializes" -  It can specialize - or - it specializes?

496 - "refter" - refer

521-522 - "infromation"

523 - "simlarity"

Nevertheless, the whole comment is that the article deserves publication, with minor corrections, such as limitations of the research and deeper discussed outcome.

Good luck, and hope to see the article published soon.

Author Response

Dear Reviewer,

We really appreciate the valuable comments and suggestions from the reviewers, which makes the paper more comprehensive.  In the following, we present a point-by-point response to the comments. The revised version of the manuscript with marked-up changes in blue color is provided.

  1. The positive impression of the material could be better, if the discussion of the results would present a deeper analysis of the outcome of the study.

Author response:  We add more analysis of the experiment results in Sec 5, and explain the reason of differences in the performance.

  1. The limitations of the research should also be mentioned, such as answers on several questions regarding the methodology (or these remarks could be enumerated as limitations of the study):
    1. If many questions concerns an issue (a concept), what is the influence of number of iterations to increase the grasp and the mastery of knowledge?

Author response:  We use context C3 to describe the relationship of the question and the mastery level of relevant knowledge concepts. For this context, we utilize QR-matrix to represent. And we combine it with student’s responses as the interaction sequence, and input them into both LSTM-based model and multi-head attention model to study how the mastery level of knowledge affects the student’s performance of the question. Our interaction sequence contains the questions that the student has answered, which share common knowledge concepts. The responses of the student for the questions contains the same knowledge concept will jointly affect the mastery of this concept, which is analyzed by both LSTM-based and multi-head attention model, so that we can utilize it to calculate its influence on the current practice and predict the student’s performance.

  1. do the authors classify the students according (1) the speed of grasping and depth of mastery of knowledge, and (2) their psychic type: the intro/extra-version, neurotism, et other features of personality? The "psychological state" described in section 4.5 is not clear.

Author response: (1) In this paper, our study focuses on tracing student's knowledge status. For the students’ performances, we measure them with the objective factor, that is, the correctness of their responses, since it is easy to directly judge whether their performances are good. In the current work of knowledge tracing, the speed of grasping and depth of mastery of knowledge are not considered, which could be good topics to study further.  (2) There are some psychological factors can affect students’ performances. Our model provides a uniform framework, which can support students with all types of personality traits. Since the limitation of data sets, we could not discuss the influence of personality traits in our paper, but we will consider them in the future work. (3) The "psychological state" is used to describe the influence of study results to the student’s positivity, which is generated from the law of effect. We rewrite this part and give an explanation for it in the paper in Sec.6.

  1. does the T and t differ from student to student, from level to level, from discipline to another? Is T and t limited, i.e., one minute for a question? Is the time to answer a question is taken into account, e.g., if a student is asnwering too long, what happens, the answer is marked as wrong? Or not, the period can be infinite?

Author response:  Each question has a limit time to answer. If a student cannot answer in time, his/her answer will be judged incorrect. We add this qualification into paper to make it more complete in Subsec 4.1.

  1. do the researchers take into account a potential fraud?

Author response:  We assume that students are honest in the practice, that all their behaviors and responses are truthful and valid. We add this assumption in the paper to make it more rigorous in Subsec 4.1.

These remarks could improve the content, giving more correct understanding of the further options to implement the outcome.

The last remark concerns many typos to correct, such as:

line 6 - "thier" - may be, their?

58 - "exmaple" - example?

75 - "inforamtion" - information?

line 81 - "affact students’ performane" - affect students’ performance?

82 - "and designd" - design?

85 - "to describes" - to describe?

87 - "fogetness" - "forgetness"

91 - "desinged" - designed?

93 - edcuational

100 - "traing" - tracing?

107 - "... and Andercon [14]" - Anderson

199 - "Meningful" - meaningful?

202 - "desgin" - design

205 - "he/her" - he/she

219 - "explicit contexts that we can represent them directly" - that we can represent directly - or - and we can represent them directly

233 - "similariy" - similarity

242 - "exercis" - exercise

294 - "influened"

361 - "similarr"

455-456 - " It can specializes" -  It can specialize - or - it specializes?

496 - "refter" - refer

521-522 - "infromation"

523 - "simlarity"

Author response:  We are sorry for our errors in expression. We corrected all these errors in the manuscript and did reading-proof carefully with our best efforts.

Round 2

Reviewer 1 Report

Satisfactory